# Neural Architecture Search via Ensemble-Based Knowledge Distillation

## Abstract

Neural Architecture Search (NAS) automatically searches for well-performed network architectures from a given search space. The One-shot NAS method improves the training efficiency by sharing weights among the possible architectures in the search space, but unfortunately suffers from insufficient parameterization of each architecture due to interferences from other architectures. Recent works attempt to alleviate the insufficient parameterization problem by knowledge distillation, which let the learning of all architectures (students) be guided by the knowledge (i.e., parameters) from a better-parameterized network (teacher), which can be either a pre-trained one (e.g., ResNet50) or some searched out networks with good accuracy performance up to now.

However, all these methods fall short in providing a sufficiently outstanding teacher, as they either depend on a pre-trained network that does not fit the NAS task the best, or the selected fitting teachers are still undertrained and inaccurate. In this paper, we take the first step to propose an ensemble-based knowledge distillation method for NAS, called EnNAS, which assembles an outstanding teacher by aggregating a set of architectures currently searched out with the most diversity (high diversity brings highly accurate ensembles); by doing so, EnNAS can deliver a high-quality knowledge distillation with outstanding teacher network (i.e., the ensemble network) all the time. Eventually, compared with existing works, on the real-world dataset ImageNet, EnNAS improved the top-1 accuracy of architectures searched out by 1.2% on average and 3.3% at most.

## 1 Introduction

Neural architecture search (NAS) (Elsken et al., 2019) can automatically search for the optimal neural architecture from a given search space. Neural architectures discovered by NAS have surpassed manually designed ones in accuracy on various tasks (Zoph & Le, 2016; Pham et al., 2018; Bender et al., 2018). Instead of parameterizing each searched architecture from scratch, one-shot NAS methods (Liu et al., 2018; Cai et al., 2018; Guo et al., 2020b) have been wildly used recently due to their moderate computational cost. One-shot NAS achieves this by letting a whole NAS search space share the parameters and form a *supernet*, and the parameterization of the supernet only requires *one* training process. Unfortunately, despite its efficiency, one-shot NAS is often denounced for its imprecise estimation of candidate architectures' quality (Bender et al., 2018; Li et al., 2020c; Zhang et al., 2020), which is evaluated by letting each candidate *subnet* directly inherit parameters of the supernet.

Several studies have attributed this imprecise estimation in one-shot NAS to the insufficient parameterization of search space, caused by two major factors: (1) interference between architectures (Li et al., 2020c) and (2) unfair training opportunities among candidates (Chu et al., 2019). Among amendments to address this insufficient parameterization issue, Knowledge Distillation (KD) (Hinton et al., 2015) is a promising technique widely used recently (Cai et al., 2019; Yu et al., 2020; Li et al., 2020a). It adopts an external teacher to regulate the supernet training to improve the parameterization efficiency , resulting in a higher quality of searched-out architecture compared with conventional one-shot methods (Guo et al., 2020b).

Unfortunately, although knowledge distillation is promising, how to select an **high-quality** teacher remains an open challenge. On known datasets and tasks, existing work typically chooses a trained,

known-to-be-good network as the teacher. DNA (Li et al., 2020a) and BigNAS (Yu et al., 2020) bring in high accuracy guidance from a trained network to perform **supervised** KD. Although the accuracy of a teacher is critical, this does not solely guarantee good teaching results (Cho & Hariharan, 2019; Mirzadeh et al., 2020).

An **unsupervised** method, Cream (Peng et al., 2020), surpassed supervised ones recently. It dynamically maintains a pool of subnets with superior accuracy as the teachers to teach the remaining ones. Cream focuses more on the consistency and complementarity between teachers and students for higher KD efficiency, which accounts for its high performance in test accuracy. But it still fails to find a high-quality teacher. On the contrary, it ignores the fundamental factor, i.e., the accuracy of teachers. The average accuracy gap between teachers and students is 2.1% in Cream, suggesting that teachers do not have much extra knowledge to teach. This inferior teacher accuracy greatly reduces the efficiency of KD and will constrain the upper bound of students' final performance. To conclude, no prior work achieves both high accuracy and good suitability when selecting the teachers.

In order to better meet the above two-fold requirement for high-quality teachers of KD one-shot NAS, we focus on improving the accuracy of teachers selected in unsupervised methods. Recent advances in ensemble online knowledge distillation (Zhang et al., 2018; Lan et al., 2018; Guo et al., 2020a) managed to train multiple networks simultaneously and efficiently via knowledge distillation without pre-training a teacher. Following them, in this paper, we propose EnNAS, an ensemble-based knowledge distillation method for NAS. Specifically, For each input batch, EnNAS samples a set of candidate subnets as students, **assembles** outputs from the students as a teacher, and performs knowledge distillation between the assembled teacher and each student. Ensemble can address the problem of low teacher accuracy in previous unsupervised works, while maintaining other good characters inherited from them, including good complementarity between teachers and students.

A challenge is how to select the set of subnets for maximum knowledge distillation effect each time, i.e., selecting a set that can be assembled into a teacher of higher accuracy to teach themselves better. Ensemble tends to yield better results when diversity presents among the outputs of networks (Kuncheva & Whitaker, 2003). Therefore, we propose a diversity-based sampling strategy to select subnets generating more diverse outputs than random sampling. Precisely, EnNAS measures the feature map similarity between operators in the same layer. An operator has a higher probability of being sampled when its similarity with other operators in the same layer is low. By doing so, the subnets generate more diverse feature maps and produce more diverse outputs. Our experiments demonstrate that the diversity-based sampling strategy managed to assemble teachers that are 5.8% more accurate than students on average.

This work makes the following contributions:

- We propose to combine unsupervised learning and ensemble learning into the training process of KD one-shot NAS. It manages to raise the accuracy of teachers while preserves beneficial factors of unsupervised learning, The teachers it employed meet the two-fold requirement for teachers in KD one-shot NAS better than ever.

- We design a diversity-based sampling strategy for EnNAS to generate high-quality ensembles. It guarantees that the selected student subnets generate diverse outputs. Based on the diverse outputs, the assembled teacher has higher accuracy than previous works. Teachers in EnNAS reach 5.8% higher accuracy than students on average, while the best previous work (Peng et al., 2020) reaches only 2.1%. This increases the upper bound of accuracy in KD one-shot NAS.

- EnNAS sets up a new SOTA in one-shot NAS on Imagenet. Extensive evaluations show that compared with existing works, EnNAS improved the accuracy of produced networks by 1.2% on average and at most 3.3% on ImageNet.

## 2 RELATED WORKS

### 2.1 ONE-SHOT NEURAL ARCHITECTURE SEARCH

Neural Architecture Search (NAS) has become the mainstream approach to automate the manual process of architecture design, which significantly reduces the substantial effort of human experts (Zoph & Le, 2016; Pham et al., 2018). Existing NAS strategies can be classified into two

categories: *sample-based* NAS and *weight-sharing* NAS. In *sample-based* NAS strategies (Zoph et al., 2018; Wang et al., 2020; Zhong et al., 2020), subnets are sampled from a huge search space with different sampling strategies (e.g., randomized sampling (Bender et al., 2018), uniform sampling (Dong & Yang, 2019)), and then fully trained on the target dataset from scratch. However, training each subnet from scratch is expensive. Hence, *sampled-based* NAS strategies only can support architecture searching within small target datasets (e.g., CIFAR-10).

On the contrary, the *weight-sharing* strategies (Khodak et al., 2019; Li et al., 2020b) (a.k.a., one-shot strategies) can greatly reduce the training cost, supporting large datasets like ImageNet. State-of-the-art *weight-sharing* NAS, such as Differentiable Architecture Search (DARTS) (Liu et al., 2018) and its variants (e.g., P-DARTS (Chen et al., 2019)), encode the search space into a *supernet*. A supernet is composed of all candidate operations (e.g., conv3x3) and maintains weights for each operation. The supernet is only trained once. All subnets sampled from the supernet can directly inherit the weights in the supernet that share common graph nodes. Thus, the qualities of subnets are directly queried on the inherited weights.

However, despite many benefits, one-shot strategies with weight sharing suffer from *weight coupling*: the operators of a supernet are jointly optimized, and from a micro view, each subnet's weight is interfered by other subnets. Thus, one-shot strategies produce subnets with low validation accuracy compared with their ground-truth modeling ability, especially on large datasets (e.g., ImageNet). To reduce such coupling, SPOS (Guo et al., 2020b) in each training step samples one subnet to be trained, and weights are still shared between subnets.

## 2.2 NEURAL ARCHITECTURE SEARCH WITH KNOWLEDGE DISTILLATION

Knowledge distillation (Hinton et al., 2015) for the neural network is to transfer the knowledge from a trained teacher to a smaller student. Existing works on knowledge distillation can be roughly classified into two categories. The first category is to use soft labels generated by the teacher to teach the student, which was first proposed by (Ba & Caruana, 2013). Hinton et al. (Hinton et al., 2015) redefined knowledge distillation as training a shallower network to approach the teacher's output after the softmax layer. However, when the teacher model gets deeper, learning the soft-labels alone is insufficient. To address this problem, the second category proposes to employ the internal representation of the teacher to guide the training of the student (Romero et al., 2014; Zhang et al., 2017; Yim et al., 2017; Wang et al., 2018).

There are already literatures that adopt knowledge distillation to accelerate NAS. Cream (Peng et al., 2020) lets each subnet, which is sampled for each training step, not only learn from the dataset, but also learn from a teacher subnet with the highest up-to-date accuracy. However, Cream still wastes learning opportunities and fails to efficiently spread the knowledge throughout the supernet. DNA (Li et al., 2020a) adopts an external teacher DNN that learns the dataset to *supervise* the NAS process, and all other NAS candidates learn from the teacher. However, the learned knowledge is highly bounded by the external teacher. In this work, EnNAS iteratively generates the teacher through the ensembling of multiple student subnets, and then the ensemble teacher guides the training of those student subnets. Thus, EnNAS's strategy is not bounded by any static teacher, and the learning process is totally automatic and *unsupervised*.

## 3 METHOD

### 3.1 OVERVIEW

We first give an overview of our proposed method EnNAS that accelerates one-shot NAS through ensembling-based knowledge distillation. In each iteration, EnNAS selects multiple subnets from the supernet and optimizes these subnets simultaneously. The ensembling soft targets are generated by aggregates logits of all subnets. All the subnets learn from the ground-truth labels as well as the soft targets. The subnets are selected by the diversity-based sampling strategy of EnNAS, which takes the operator diversity into consideration.

## 3.2 ONLINE DISTILLATION

Suppose we have selected $m$ subnets from the supernet for a given input batch $\{x, y\}$, where $x$ is the input sample and $y$ is the ground-truth label, the $i-th$ subnet outputs logit $z_i$ and produces soft probability distribution $q_i = softmax(z_i)$. The logit $t$ of the ensemble teacher is a weighted sum of all the subnet logits

$$t = \sum_{i=0}^{m-1} w_i z_i, \text{ subject to } \sum_{i=0}^{m-1} w_i = 1 \tag{1}$$

where $w_i$ is the weight corresponds to logit of the $i-th$ subnet. Following the ensembling method proposed in Guo et al. (2020a), the teacher logit is calculated by minimizing the cross-entropy loss of the ensemble teacher logit $t$ and the ground-truth label $y$. Let $w = \{w_0, w_1, \cdots, w_{m-1}\}$ and $Z = \{z_0^T; z_1^T; \cdots; z_{m-1}^T\}$, the problem is formalized as

$$\min_{w} \mathcal{L}_{CE}(w^T Z, y) \tag{2}$$

The problem is convex and we solve it using the ECOS solver (Domahidi et al., 2013).

After generating the ensemble teacher logit, we can get the teacher soften probability distribution $p = softmax(t)$. The loss $\mathcal{L}_i$ of the $i-th$ subnet is then a combination of the cross-entropy loss with the ground-truth label $y$ and the knowledge distillation loss with the soft target $p$.

$$\mathcal{L}_i = \mathcal{L}_{CE} + \mathcal{L}_{KD}, \text{ where } \mathcal{L}_{KD} = -\tau^2 KL(p, q) \tag{3}$$

The knowledge distillation loss bases on the KL divergence between $p$ and $q$. $\tau$ and $\lambda$ are the hyperparameters. Each subnet uses the combined loss $\mathcal{L}$ to update the weights of corresponding operators in the supernet.

## 3.3 DIVERSITY-BASED SAMPLING

In this section, we introduce the diversity-based sampling strategy of EnNAS. This strategy lets EnNAS select subnets that generate more diverse logits, which helps to generate a more accurate ensemble teacher logit. We first define the concept of operator diversity and how to calculate it, and then we explain the diversity-based sampling strategy based on the operator diversity.

We assume the supernet $\mathcal{N}$ has $L$ layers, and each layer $\mathcal{N}^l$ $(0 \leq l \leq L-1)$ has $T$ candidate operators denote as $\{o_0, o_1, \cdots, o_{T-1}\}$.

The operator diversity is defined as the diversity of feature maps output by two operators in the same layer regarding the same input. The input of an operator is the feature maps from the operator's last layer. Formally, given all possible input feature maps $\mathcal{U} = \{u_0, u_1, \cdots, u_{n-1}\}$ in layer $\mathcal{N}^l$, the diversity of two operators $o_m$ and $o_n$ in the layer is defined as:

$$\zeta(o_m, o_n) = 1 - \frac{\sum_{i=1}^{n} cos(r(o_m(u_i)), r(o_n(u_i)))}{n} \tag{4}$$

where $r(\cdot)$ is a function that reshapes a feature map to a vector. In general, the value of $\zeta(o_m, o_n)$ lies between 0 and 1. The large $\zeta(o_m, o_n)$ value corresponds to high diversity between operators $o_m$ and $o_n$.

However, it is infeasible to get all the possible input feature maps for a layer in the supernet, and we approximate the operator diversity by sampling a subset of the possible input feature maps. Specifically, we sample inputs from the dataset and feed them into the supernet. The feature maps input to a layer are uniformly assigned to all operators in that layer, and the output feature maps of all operators are merged as the input feature maps to the next layer. By doing so, we get the sampled input feature maps of each layer.

Next, we describe the diversity-based sampling strategy of EnNAS. The main idea is to sample operators that generate more diverse feature maps to form subnets. Since we sample $m$ subnets for an input batch, $m$ operators are selected from each layer in the supernet. For a layer in the supernet that contains operators $\{o_0, o_1, \cdots, o_{T-1}\}$, we denote the $m$ operators sampled as

$\{x_0, x_1, \cdots, x_{m-1}\}$. We sample the operators sequentially, and the probability of sampling an operator is based on its diversity with other operators already sampled. Specifically, for an operator set $\mathcal{A} = \{\tilde{x}_0, \tilde{x}_1, \cdots, \tilde{x}_{h-1}\}$ which contains $h$ sampled operators, the probability of sampling operator $o_i$ ($0 \le i \le T - 1$) as the next operator is defined as

$$p(o_i|\mathcal{A}) = \frac{g(o_i, \mathcal{A})}{T \sum_{\mathcal{A}' \in \mathbb{A}} g(o_i, \mathcal{A}')}, \text{ where } g(o_i, \mathcal{A}) = \frac{\sum_{o_j \in \mathcal{A}} \zeta(o_i, o_j)}{size(\mathcal{A})} \tag{5}$$

$\mathbb{A}$ is a set that contains all possible choices of sampling $h$ operators. $g(o_i, \mathcal{A})$ measures the average diversity between operator $o_i$ and the already sampled operator set $\mathcal{A}$. The probability of sampling an operator increases if it generates feature maps more diverse from the operators already sampled. Besides, the probability of choosing any operator as the next operator equals $\frac{1}{T}$.

To sum up, the diversity-based sampling strategy in EnNAS has the following two properties:

- The subnets sampled generate more diverse outputs than random sampling.
- The probability of choosing each operator is the same, and thus all the operators are trained equally.

### 3.4 NAS WITH ONLINE DISTILLATION

**Training.** Given an input batch, we sample $m$ subnets based on the probability distribution derived above. The loss of each subnet is computed based on the loss function described in Section 3.2, and backpropagation is performed using the loss. All subnets update weights of the operators they contain by gradient descent.

After training the supernet for several iterations, we follow the process described in Section 3.3 to update the probability distribution of operators in the supernet correspondingly. Since the operator diversity does not change drastically, updating operator diversity per epoch does not affect the diversity of subnets sampled.

**Searching.** After the supernet is trained, we perform an evolution search on it to get an architecture with optimal accuracy on the validation dataset. We first randomly initialize the evolution search with $n$ architectures randomly picked. We then evaluate the accuracy of each architecture on the validation set. The $k$ architectures with the highest accuracy are picked in each generation. The picked architectures served as parents to generate the next generation by crossover and mutation. After a few iterations, the architecture with the highest accuracy is selected.

---

**Algorithm 1** NAS with Online Knowledge Distillation

---

**Require:** supernet $\mathcal{A}$ w/ initialized weights $w$, dataset $\mathcal{D}$ w/ batches $\{(X^{(i)}, y^{(i)})\}_{i=1}^n$
 1: **while** $\mathcal{A}$ not converged **do**
 2:     select teacher $\alpha_t$ based on generalization error
 3:     sample a batch $(X^{(i)}, y^{(i)})$ from dataset
 4:     sample $m$ subnets $(t_0, \cdots, t_{m-1})$
 5:     **for** $i = 0$ **to** $m - 1$ **do**
 6:         calculate logit $z_i$ of subnet $t_i$ with respect to $X^{(i)}$
 7:     **end for**
 8:     calculate ensemble logit $p$ based on $(z_0, \cdots, z_{m-1})$
 9:     **for** $i = 0$ **to** $m - 1$ **do**
10:         calculate $\mathcal{L}_{CE}$ and $\mathcal{L}_{KD}$ for $t_i$
11:         update $t_i$ weights by $\nabla_w \mathcal{L}_{CE}$ and $\nabla_w \mathcal{L}_{KD}$
12:     **end for**
13: **end while**
14: Derive the architecture from the supernet

---

## 4 EVALUATION

**Datasets.** We fully implemented our method EnNAS and used EnNAS to automatically find the most accurate architectures for image classification tasks. Our experiments were conducted on a popular

Table 1: Comparison with state-of-the-art NAS methods on ImageNet. EnNAS w/o DS means EnNAS without diversity-based sampling.

| Space | Method | Top-1 (%) | Top-5 (%) | FLOPs (M) | Supervised | Supernet Training (GPU days) |
|---|---|---|---|---|---|---|
| | SPOS | 74.7 | - | 328M | N | 12 |
| | Cream | 77.6 | 93.3 | 287M | N | 16 |
| s1 | DNA | 77.1 | 93.3 | 348M | Y | 4 |
| | EnNAS w/o DS | 77.5 | 93.5 | 317M | N | 12 |
| | EnNAS | **77.9** | **93.6** | 330M | N | 12 |
| | SPOS | 75.1 | 92.5 | 479M | N | 14 |
| | Cream | 77.3 | 93.0 | 422M | N | 19 |
| s2 | DNA | 77.6 | 93.1 | 396M | Y | 5 |
| | EnNAS w/o DS | 78.0 | 93.6 | 436M | N | 14 |
| | EnNAS | **78.4** | **93.8** | 413M | N | 14 |
| | SPOS | 77.1 | 93.3 | 482M | N | 17 |
| | Cream | 78.7 | 93.7 | 495M | N | 22 |
| s3 | DNA | 78.2 | 93.2 | 471M | Y | 7 |
| | EnNAS w/o DS | 78.6 | 93.4 | 519M | N | 17 |
| | EnNAS | **79.1** | **94.2** | 532M | N | 17 |

image classification dataset, i.e., ImageNet Deng et al. (2009). ImageNet contains over 14 million images, and specifically, we used ILSVRC2012 Russakovsky et al. (2015), a subset of ImageNet which contains 1,000 object classes and 1.28M training images and 50K validation images.

**Baselines.** We compared EnNAS with three NAS methods that achieve state-of-the-art performance results, including SPOS Guo et al. (2020b), DNA Li et al. (2020a), and Cream Peng et al. (2020). SPOS is the commonly-used one-shot approach that randomly samples a subnet and trains it using a single batch iteratively. DNA uses a pre-trained network as the teacher and distills it to the student supernet use block-wise knowledge distillation. Cream leverages knowledge distillation to accelerate NAS by maintaining a teacher pool and select a teacher from the pool for each subset sampled. We also add a strawman approach that integrates online-ensembling to one-shot NAS through randomly sampling a set of subnets for each input batch, namely EnNAS w/o diversity-based sampling.

**Search.** We benchmark EnNAS on three search spaces. The first search space *s1* is a **small** search space similar to state-of-the-art works for a fair comparison. The search space *s1* contains a set of mobile inverted bottleneck convolution (MBConv) with kernel sizes {3, 5, 7} and expand ratios {1, 3, 6}. An Identity operator is also added for elastic depth search. We further use two **large** search spaces *s2* and *s3* to evaluate EnNAS in a more complicated search space. We add the building blocks of ShuffleNetV2 (Ma et al., 2018) into search space *s1* to construct search space *s2*. We also increase the layers in search space *s2* to construct search space *s3*.

**Settings.** Our experiments were conducted on the 8 Nvidia 2080TI GPUs, and each GPU is equipped with 11 GB physical memory. We train the supernet for 120 epochs using the following settings: SGD optimizer with momentum 0.9 and weight decay 4e-5, initial learning rate 0.5 with linear annealing. The chosen architecture is retrained for 500 epochs on Imagenet using an RMSProp optimizer with initial learning rate 0.064 and weight decay 1e-5. Both EnNAS and Cream start knowledge distillation from the 20th epoch. EnNAS samples 4 subnets for each input batch. EnNAS samples only $\frac{1}{4}$ of the training dataset in each epoch for a fair comparison with the other baselines.

## 4.1 END-TO-END EVALUATION

**Metrics.** We leveraged three commonly-used metrics to evaluate different NAS techniques' end-to-end performance, including *Top-1/5 Accuracy*, *FLOPs* (i.e., floating point operation number of an architecture) and *Search Cost*. Among these metrics, *Top-1/5 Accuracy* stands for the accuracy of searched architecture on the test dataset.

**Results.** We show the results compared with state-of-the-art NAS methods on ImageNet in Table 1. We re-implemented the baseline methods (i.e., Cream, DNA, and SPOS) on search space *s2* and *s3* since they have not been evaluated on these search spaces. EnNAS achieved an average top-1 accuracy of 78.4% on three search spaces, which is 1.2% higher than the average top-1 accuracy compared to state-of-the-art methods. This shows the effectiveness of our knowledge

distillation method, which effectively transfers knowledge from the well-performed subnets to the poorly-performed subnets. Besides, our method takes the same GPU days as SPOS to complete the architecture search. This indicates the knowledge distillation mechanism does not incur extra overhead.

We can also observe that although Cream improves the architecture accuracy compared with SPOS, it incurs more training cost (i.e., 4 to 6 days on ImageNet). This is because, for each subnet sampled to be trained, Cream has to calculate the logits of all teachers in the teacher pool. After that, only one logit output is used for distillation. Thus, the training efficiency of Cream degrades largely. EnNAS, on the other side, leverages an ensemble teacher to train all the student subnet. All logits of the teachers contribute to the NAS process.

## 4.2 ANALYSIS OF SEARCH PROCESS

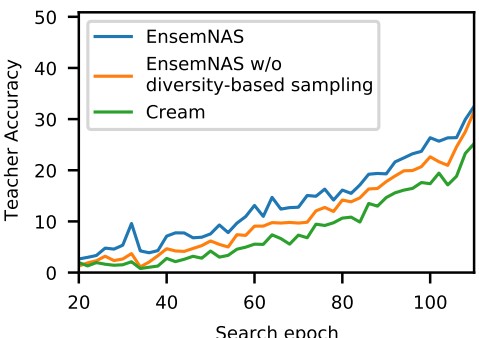

Figure 1: Comparison of teacher accuracy of 3 methods

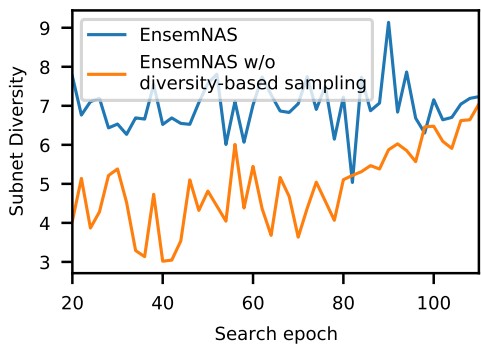

Figure 2: Diversity of subnets sampled of 2 methods

To analyze the effectiveness of our method, we compared teacher top-1 accuracy during the training process of EnNAS, EnNAS w/o diversity-based sampling, and Cream. We randomly sampled 100 teachers used by those methods every epoch. We get the accuracy of these teachers using the validation set of ImageNet. Figure 1 shows the average accuracy of sampled subnets of 3 methods. EnNAS achieves better average accuracy than Cream starting from the 20 *th* epoch. At the end of supernet training, the average teacher accuracy of EnNAS is 4.5% higher than that of Cream. Besides, compared with EnNAS w/o diversity-based sampling, EnNASalso achieves higher teacher accuracy during the NAS process. This is because EnNAS selects subnets generate more diverse outputs to form a better ensemble teacher. We also evaluated the accuracy gap between teacher and student subnets for the 3 methods. The teachers of Cream are 2.1% higher than the subnets as students, while the ensemble teachers of EnNAS achieve 5.8% higher accuracy than student subnets.

Both EnNAS and EnNAS w/o diversity-based sampling generate teachers with higher accuracy because of the online ensebmling technique in EnNAS. The teachers in EnNAS promote the convergence of subnets in the supernet due to the larger accuracy gap.

## 4.3 ABLATION STUDY

In this subsection, we evaluate the effectiveness of the diversity-based sampling strategy in EnNAS. We measure the diversity of subnets sampled using both the diversity-based sampling strategy and random sampling. The diversity is measured by the average Euclidean distance between the logits of each pair of subnets. As shown in Figure 2, the subnet diversity selected by EnNAS is larger than randomly sampling (i.e., EnNAS w/o diversity-based sampling). This demonstrates that the diversity-based sampling strategy proposed by EnNAS successfully selects subnets that generate more diverse outputs. To show whether diversity among subnets leads to a better ensemble teacher, we also evaluated the accuracy of ensemble teachers generated by those two methods, as discussed in Section 4.2. Evaluation shows that the ensemble teachers of EnNAS are 2.5% higher than EnNAS w/o diversity-based sampling on average.

## 5 CONCLUSION

In this work, we propose an efficient neural architecture search method (i.e., EnNAS) that leverages knowledge distillation. We observe that the previous NAS methods ignore the teacher selection problem existed in NAS, which can mistakenly use teacher that leads to low distillation efficiency. We solve this problem by generating the teacher through online ensebmling. We then incorporate the ensemble teacher into our workflow. Our evaluation results show that EnNAS can out-perform existing NAS methods with better accuracy.

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
