# OpenReview forum: "Neural Architecture Search via Ensemble-based Knowledge Distillation"
_ICLR.cc/2022/Conference — ICLR 2022 Submitted_

### Official Review · Reviewer_4GyZ · 2021-11-02

**Correctness:** 3
**Technical Novelty And Significance:** 2
**Empirical Novelty And Significance:** 2
**Recommendation:** 3
**Confidence:** 4

**Main Review:**

Strengths:
1. The proposed method avoids pre-selecting a sufficiently accurate teacher network to guide the training of super-net. Instead, the teacher network is dynamically constructed and will have higher accuracy as the super-net is gradually trained to its optimum.
2. The diversity-based criterion for selecting student networks seems to be a reasonable choice for constructing a high-quality ensemble as the teacher network.

Weaknesses:
1. Comparison with some more recent weight-sharing NAS methods are missing. For instance, AlphaNet [1] also aims at improving the training of super-net by replacing the commonly used KL-divergence with alpha-divergence in knowledge distillation. It seems that they achieved a better accuracy-FLOPs trade-off than this paper (~79% @ 300 MFLOPs and ~80% @ 500 MFLOPs).
2. For constructing the ensemble, student networks are sampled based on the diversity criterion. However, the prediction accuracy of each student network is not considered. Will this lead to a sub-optimal teacher network, if the search space is not well designed (e.g. contains many low-quality operator choices)?
3. Section 3.2, last paragraph. What does $\lambda$ stand for? Besides, why there is a negative sign in the knowledge distillation loss in Equation (3)?
4. Section 3.3, Equation (4). The cosine similarity ranges from -1 (completely opposite) to 1 (identical), which means that the diversity metric defined here ranges from 0 to 2, instead of "between 0 and 1".

[1] Dilin Wang, Chengyue Gong, Meng Li, Qiang Liu, Vikas Chandra. AlphaNet: Improved Training of Supernets with Alpha-Divergence. ICML 2021.

**Summary Of The Paper:**

In this paper, authors propose to dynamically construct a teacher network from the ensemble of multiple student networks, and then use it to guide the training of super-net via knowledge distillation. The ensemble of student networks is built with the diversity criterion explicitly considered to improve the teacher network's accuracy. After the super-net is well trained, the optimal network architecture is determined by evolution search.

**Summary Of The Review:**

My major concerns on this paper includes:
1. The comparison between some recent weight-sharing NAS methods, e.g. AlphaNet, is missing.
2. The diversity-based sampling method may impose additional restraints of the design of search space of network architectures.

---

### Official Review · Reviewer_JuuW · 2021-11-06

**Correctness:** 2
**Technical Novelty And Significance:** 2
**Empirical Novelty And Significance:** 2
**Recommendation:** 5
**Confidence:** 5

**Main Review:**

- Strength
1. Well written. Easy to understand.
2. The motivation is clear.


- Weakness
1. Inconsistent experiment results. The authors claimed the proposed method ``achieved 1.2% higher than the average top-1 accuracy compared to state-of-the-art methods''. However, I checked Table 1 and found there is only about 0.5% higher than Cream. Also, I checked the revision history and found the authors claimed a 1.71% improvement on their first version.

2. The idea is kind of straightforward. Using the ensemble teacher has been widely used in previous knowledge distillation works and semi-supervised learning methods (e.g., mean-teacher, co-training). As for the diversity-based sampling strategy, it is quite trivial of the current design.

3. For the diversity-based sampling strategy, the authors sample operators sequentially, which might not be a good choice. In this way, the first selected/sampled operator(s) would have a huge impact on the following sampling process, which doesn't make sense. This may greatly  affect the final performance since the sampled set would be totally different given different random initializations/seeds.

4. The authors did not try other sampling strategies as baselines. For example, always choosing subnets with most different connections might be a strong baseline.

5. What is \lambda just below Eq. (3)? Is it a typo or something I missed.

6. What is the impact of the hyperparameter \tau? Is it very sensitive?




**Summary Of The Paper:**

This paper focuses on knowledge distillation for NAS, which improves the student networks by the ensemble teacher models. To generate better ensemble teacher logits, the authors proposed to sample operators that predict diverse feature maps to form subnets.
Experiments shows the proposed method achieves better results compared to state-of-the-art KD-based NAS methods.

**Summary Of The Review:**

The solution is somehow trivial (e.g., diversity-based sampling strategy). The experiments are not sufficient.

---

### Official Review · Reviewer_72m4 · 2021-11-06

**Correctness:** 2
**Technical Novelty And Significance:** 3
**Empirical Novelty And Significance:** 2
**Recommendation:** 3
**Confidence:** 4

**Main Review:**

Strengths:
- This paper explores and tackles an useful area of improving the teacher network in self-distillation augmented NAS.
- They propose a concrete algorithm to create stronger teachers using ensembling of subnetworks and and demonstrate that they can achieve higher performance on Imagenet compared to similar methods in a several albeit similar search spaces.
- They introduce a novel diversity sampling algorithm to improve the diversity of models used to build the ensemble and demonstrate its affects with ablations. They show that it it improves final model performance and allows the construction of a stronger teacher with less training.

Weaknesses:
- While they achieved significant results on Imagenet, the paper could be significantly improved by inclusion of significantly more empirical results. It would be especially useful to demonstrate that the one-shot network can still transfer to other datasets. It would also benefit from results on the NAS benchmark datasets to better characterize the ability of the algorithm in generally characterizing the architecture space. Are the results you show from only one run of the algorithms?
- The current wording and presentation of the paper make important details of the paper and the algorithm difficult to follow. It is unclear if all of the claims are strongly supported.
- The algorithm itself could be much better understood with significantly more ablations of the different hyperparameters like the number of subnets chosen, and the weighting between the distillation loss and the Cross entropy loss.



Questions:
In the abstract and results you claim, "Eventually, compared with existing works, on
the real-world dataset ImageNet, EnNAS improved the top-1 accuracy of architectures searched out by 1.2% on average and 3.3% at most."  I'm not sure based on table 1 compared to CREAM if that is accurate? I believe CREAMS average = 77.86% EnNAS average = 78.47%

Will the code be released for reproducibility?

"After training the supernet for several iterations, we follow the process described in Section 3.3 to update the probability distribution of operators in the supernet correspondingly. Since the operator diversity does not change drastically, updating operator diversity per epoch does not affect the diversity of subnets sampled."

What are the details for updating the probability distribution? It does not seem to appear in the algorithm psuedocode. Is it updated per epoch? Since the distribution is dependent on the previously selected subnets, how doe that work? Does this bias the one-shot network towards a particular architecture distribution? How many images are used to determine this diversity sampling distribution?  In section 3.3, you claim that the method both biases toward diverse outputs and every operator is trained equally. Could you explain further? I believe I am misunderstanding.

Is this method unsupervised? The paper seems to claim that this paper, and the CREAM paper are unsupervised, but I believe both train on the labels of Imagenet. It might be a bit confusing to overload the meaning of that term. Calling it self-distillation may be less confusing?

Looking at Figure 1 and 2, it seems like the gap in performance of the algorithms and diversity of subnets with and without diversity sampling significantly drops towards the end of training? Would they converge if you continued training longer? How many runs did you do in the abaltions

Does there happen to be a particular theoretical grounding for the specific derivation of the diversity-based sampling based being linearly proportional to the cosine similarity?

Minor:
ensebmling in the conclusion

**Summary Of The Paper:**

This paper proposes an efficient concrete novel Neural Architecture KD one-shot NAS algorithm that uses a diversity-based sampling algorithm to assemble outstanding ensemble-based teachers for Knowledge Distillation. They demonstrate their method on Imagenet with several different search spaces and significantly improve the SOTA accuracy for KD one-shot NAS. They demonstrate in ablation studies that the diversity sampling algorithm improves both the accuracy of the teacher network used in distillation as well as the performance of the final derived architectures.

**Summary Of The Review:**

This paper explores a useful area of improving the performance of the teacher network in self-distillation augmented NAS and shows that this can empirically improve NAS performance. Unfortunately, I currently recommend rejection of this paper since it does not have sufficient empirical results to demonstrate the robustness of the proposed improvement and the robustness of their method on different datasets and as a one-shot method. The paper's presentation is also unclear and some of the results may be stated incorrectly so it is unclear if all the conclusions are well justified.

---

### Decision · Program_Chairs · 2022-01-20

**Decision:**

Reject

**Comment:**

All reviewers recommend rejection, and I'm following this recommendation.